# Genetic ancestry, skin pigmentation, and the risk of cutaneous squamous cell carcinoma in Hispanic/Latino and non-Hispanic white populations

Eric Jorgenson [1,6], Hélène Choquet [1,6✉], Jie Yin[1], Thomas J. Hoffmann [2,3], Yambazi Banda[2], Mark N. Kvale[2], Neil Risch[1,2,3], Catherine Schaefer [1] & Maryam M. Asgari [4,5]

Although cutaneous squamous cell carcinoma (cSCC) is one of the most common malignancies in individuals of European ancestry, the incidence of cSCC in Hispanic/Latinos is also increasing. cSCC has both a genetic and environmental etiology. Here, we examine the role of genetic ancestry, skin pigmentation, and sun exposure in Hispanic/Latinos and non-Hispanic whites on cSCC risk. We observe an increased cSCC risk with greater European ancestry ($P = 1.27 \times 10^{-42}$) within Hispanic/Latinos and with greater northern ($P = 2.38 \times 10^{-65}$) and western ($P = 2.28 \times 10^{-49}$) European ancestry within non-Hispanic whites. These associations are significantly, but not completely, attenuated after considering skin pigmentation-associated loci, history of actinic keratosis, and sun-protected versus sun-exposed anatomical sites. We also report an association of the well-known pigment variant *Ala111Thr* (rs1426654) at *SLC24A5* with cSCC in Hispanic/Latinos. These findings demonstrate a strong correlation of northwestern European genetic ancestry with cSCC risk in both Hispanic/Latinos and non-Hispanic whites, largely but not entirely mediated through its impact on skin pigmentation.

[1] Kaiser Permanente Northern California, Division of Research, Oakland, CA 94612, USA. [2] Institute for Human Genetics, UCSF, San Francisco, CA 94143, USA. [3] Department of Epidemiology and Biostatistics, UCSF, San Francisco, CA 94158, USA. [4] Department of Dermatology, Massachusetts General Hospital, Boston, MA 02114, USA. [5] Department of Population Medicine, Harvard Pilgrim Health Care Institute, Harvard Medical School, Boston, MA, USA. [6] These authors contributed equally: Eric Jorgenson, Hélène Choquet. ✉email: Helene.Choquet@kp.org

Cutaneous squamous cell carcinoma (cSCC) is one of the most common malignancies among individuals of European ancestry, and, in recent years, its incidence has been increasing not only in non-Hispanic whites but also in Hispanics and Asians[1–5]. However, a thorough analysis of cSCC susceptibility across different ethnic groups is lacking, and data on clinical characteristics of cSCC in patients of Hispanic, Asian, or African ancestry are limited[6,7].

Lighter skin pigmentation has been shown to be a strong determinant of cSCC risk because it provides less protection against damage from ultraviolet radiation (UVR) exposure compared to darker skin pigmentation[8,9]. For example, individuals from northern Europe, who are more likely to manifest light skin color, have higher incidence rates of cSCC as compared to their southern European counterparts, who have a higher prevalence of darker skin color[10]. Genetic variation is an important contributor to variation in skin pigmentation and is also a risk factor for cSCC[11,12]. Variants in a number of pigmentation-related genes, including *SLC45A2*, *IRF4*, *TYR*, *ASIP*, *OCA2*, *HERC2*, and *MC1R*, have been associated with the risk of cSCC in populations of European ancestry[13–19], supporting the presumed biological importance of lighter pigmentation in susceptibility to cSCC. To our knowledge, the role of skin pigmentation and the effect of skin pigmentation loci on cSCC susceptibility has not been investigated in other populations.

Genetic ancestry studies have shown that the correlation between allele frequency variation and geographic distances enables the identification of an individual's geographic/ethnic ancestry[20–22]. By estimating the genetic ancestry of individuals, it is possible to determine whether geographic variation in disease prevalence, such as that observed for cSCC in European and Hispanic/Latino populations, is correlated with genetic ancestry. Variation in skin pigmentation has also been associated with genetic ancestry[23], which could explain variation in the risk of cSCC both within and between European descent populations and other populations. To date, no study has examined the role of genetic ancestry and the risk of cSCC.

To address these gaps, we conduct genetic ancestry analyses of cSCC susceptibility in Hispanic/Latino and non-Hispanic white ethnic groups from the large multiethnic Genetic Epidemiology Research in Adult Health and Aging (GERA) cohort. Next, we determine whether genetic ancestry associations are entirely due to genetically predicted skin pigmentation using skin pigmentation probabilities generated by the HIrisPlex-S program[24–26] based on 36 known skin pigment genetic variants. We then determine whether genetic ancestry associations are modified by sun exposure by considering actinic keratosis (AK) diagnosed by a physician as a clinical marker for chronic UVR exposure and assessing whether observed genetic ancestry associations are comparable at sun-protected anatomical sites and with non-cutaneous SCC (ncSCC).

## Results

### GERA cohort and cSCC risk
The study sample consisted of 11,396 cSCC cases and 86,186 controls from four ethnic groups (non-Hispanic whites, Hispanic/Latinos, East Asians, and African-Americans) (Table 1) in the GERA cohort. Among GERA participants, cSCC cases were more likely to be male (54.8%) and non-Hispanic white (96.8%). The prevalence of cSCC was much higher among non-Hispanic whites (14.0%) compared to Hispanic/Latinos (3.5%), East Asians (0.8%), and African-Americans (0.4%). Because of the limited number of cases among East Asians and African-Americans, we focused on the non-Hispanic white and the Hispanic/Latino GERA subjects for the subsequent study analyses.

**Table 1 Characteristics of the cSCC cases and controls from GERA cohort.**

|  | cSCC cases | cSCC controls |
|---|---|---|
| Age at specimen (years), mean ± SD | 70.4 ± 9.6 | 59.8 ± 13.7 |
| N (proportion that are cases) | 11,396 (11.7%) | 86,186 |
| Sex |  |  |
| Female | 5149 (8.9%) | 52,488 |
| Male | 6247 (15.6%) | 33,698 |
| Ethnicity |  |  |
| Non-Hispanic White | 11,028 (14.0%) | 67,555 |
| Hispanic/Latino | 294 (3.5%) | 8196 |
| East Asian | 61 (0.8%) | 7292 |
| African American | 13 (0.4%) | 3143 |
| Prior AK |  |  |
| Yes | 7461 (32.3%) | 15,624 |
| No | 3935 (5.3%) | 70,562 |
| Tumor location (N patients) |  |  |
| Sun-exposed only | 9996 | — |
| Sun-protected only | 420 | — |
| Both | 465 | — |

### Genetic ancestry and cSCC risk
We examined genome-wide genetic ancestry using principal components (PCs) that were calculated within each race/ethnicity group separately. Within Hispanic/Latinos, the first two PCs were geographically interpretable, with PC1 representing greater European versus Native American ancestry and PC2 representing greater African versus European ancestry[21]. Both PC1 and PC2 were associated with cSCC risk within Hispanic/Latino GERA participants, as illustrated in Fig. 1a. We identified significant associations between cSCC risk and PC1 (Model 1: $\beta = 122.48$, $P = 1.27 \times 10^{-42}$), as well as PC2 ($\beta = 43.52$, $P = 6.44 \times 10^{-3}$) (Table 2 and Supplementary Table 1), indicating that individuals with greater European ancestry had a higher risk of cSCC compared to those with greater Native American and/or African ancestry.

Within non-Hispanic whites, the first two PCs also represented geographically interpretable genetic ancestry, with PC1 characterizing a northwestern versus southeastern European cline and PC2 a northeastern versus southwestern European cline[21]. Both PC1 and PC2 were strongly associated with cSCC risk within non-Hispanic white GERA participants, as illustrated in Fig. 1b. We identified significant associations between cSCC risk and PC1 (Model 1: $\beta = 30.12$, $P = 2.38 \times 10^{-65}$), as well as PC2 ($\beta = -25.23$, $P = 2.28 \times 10^{-49}$) (Table 2 and Supplementary Table 2), indicating that individuals of northwestern European ancestry had the highest cSCC risk.

### Genetic ancestry and skin pigmentation
To determine the degree to which genetic ancestry associations with cSCC were due to genetically determined skin pigmentation, we predicted skin color for each study participant using HIrisPlex-S, a program that utilizes 36 known skin pigmentation genetic variants (Supplementary Table 3) to generate probabilities of different pigment traits[24–26]. Out of the 36 single-nucleotide polymorphisms (SNPs) that have been reported to be associated with skin pigmentation, we excluded from further analysis four SNPs (rs3212355, rs1805006, rs11547464, and rs1110400) that were not polymorphic (minor allele frequency (MAF) < 1.0%) in either GERA non-Hispanic whites or Hispanic/Latinos.

The genetic ancestry associations with cSCC were strongly but incompletely attenuated after including HIrisPlex-S skin pigmentation predictions (Model 2: $\beta = 67.31$, $P = 1.77 \times 10^{-9}$ for PC1 and $\beta = 28.43$, $P = 0.072$ for PC2 in Hispanic/Latinos; $\beta = 16.53$,

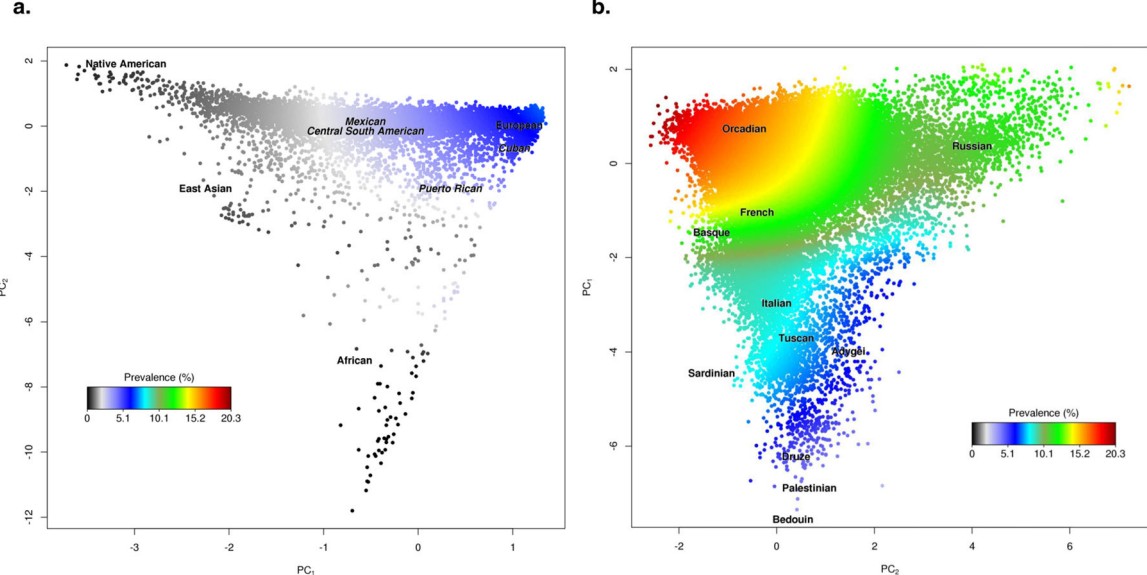

**Fig. 1 Ancestry contour figure showing cSCC prevalence by genetic ancestry in GERA. a** Hispanic/Latinos ($N = 8490$ individuals, including 294 cSCC cases); **b** non-Hispanic whites ($N = 78,583$ individuals, including 11,028 cSCC cases). cSCC prevalence is indicated on a color scale, with red hues indicating higher prevalence. Axes reflect the first two principal components of ancestry. Nationality subgroup labels were derived from the Human Genome Diversity Project.

### Table 2 Associations between genetic ancestry and cSCC risk in GERA non-Hispanic whites and Hispanic/Latinos.

| | Non-Hispanic whites | | Hispanic/Latinos | |
|---|---|---|---|---|
| | $\beta$ (SE) | *P* value | $\beta$ (SE) | *P* value |
| Model 1: Ancestry (age, sex, and PCs as covariates) | | | | |
| PC1 | 30.12 (1.76) | $2.38 \times 10^{-65}$ | 122.48 (8.95) | $1.27 \times 10^{-42}$ |
| PC2 | −25.23 (1.71) | $2.28 \times 10^{-49}$ | 43.52 (15.97) | $6.44 \times 10^{-3}$ |
| Model 2: Model 1 and HIrisPlex-S skin pigmentation predictions | | | | |
| PC1 | 16.53 (1.84) | $2.42 \times 10^{-19}$ | 67.31 (11.19) | $1.77 \times 10^{-9}$ |
| PC2 | −19.36 (1.73) | $5.33 \times 10^{-29}$ | 28.43 (15.80) | 0.072 |
| Intermediate skin | −1.32 (0.05) | $6.97 \times 10^{-150}$ | −2.23 (0.31) | $1.54 \times 10^{-12}$ |
| Dark skin | −2.24 (0.17) | $3.47 \times 10^{-41}$ | −3.22 (0.39) | $1.09 \times 10^{-16}$ |
| Model 3: Model 2 and genetic risk score | | | | |
| PC1 | 17.20 (1.84) | $1.07 \times 10^{-20}$ | 56.54 (11.49) | $8.54 \times 10^{-7}$ |
| PC2 | −18.33 (1.75) | $4.41 \times 10^{-26}$ | 28.44 (15.63) | 0.069 |
| Intermediate skin | −0.68 (0.06) | $5.94 \times 10^{-26}$ | −1.31 (0.37) | $4.54 \times 10^{-4}$ |
| Dark skin | −1.46 (0.17) | $4.44 \times 10^{-17}$ | −2.14 (0.46) | $2.93 \times 10^{-6}$ |
| GRS | 0.09 (0.01) | $8.50 \times 10^{-54}$ | 0.13 (0.03) | $1.26 \times 10^{-5}$ |
| Model 4: Model 3 and sun exposure | | | | |
| PC1 | 11.49 (1.90) | $1.52 \times 10^{-9}$ | 41.30 (11.98) | $5.67 \times 10^{-4}$ |
| PC2 | −12.99 (1.80) | $5.37 \times 10^{-13}$ | 18.67 (15.17) | 0.22 |
| Intermediate skin | −0.45 (0.07) | $1.26 \times 10^{-11}$ | −0.94 (0.39) | 0.016 |
| Dark skin | −0.96 (0.18) | $4.84 \times 10^{-8}$ | −1.62 (0.47) | $5.54 \times 10^{-4}$ |
| GRS | 0.07 (0.01) | $1.14 \times 10^{-29}$ | 0.12 (0.03) | $6.84 \times 10^{-5}$ |
| Prior AK | 1.39 (0.02) | $<2.23 \times 10^{-308}$ | 1.64 (0.14) | $5.12 \times 10^{-30}$ |

*Note*: In Models 2, 3, and 4, the HIrisPlex-S prediction for the combination of very pale skin with pale skin served as the reference group. Each model was adjusted for age, sex, and additional PCs. We also included the percentage of Ashkenazi (ASHK) ancestry as a covariate for the non-Hispanic white analyses.
*PC* principal component, $\beta$ beta, *SE* standard error, *GRS* genetic risk score (based on 14 SNPs previously reported to be associated with cSCC risk), *AK* actinic keratosis.

$P = 2.42 \times 10^{-19}$ for PC1 and $\beta = -19.36$, $P = 5.33 \times 10^{-29}$ for PC2 in non-Hispanic whites) (Table 2 and Supplementary Tables 1 and 2). Within Hispanic/Latino, dark/dark to black skin color showed strong variations in prevalence by geographic ancestry. Within non-Hispanic whites, we also observed variation in skin pigment by geographic ancestry, with lighter skin more prevalent in the northwest of Europe and dark skin more prevalent in the southeast of Europe (Supplementary Figs. 1 and 2).

Many pigmentation SNPs (17/30 = 56.7%) were associated with cSCC risk in the GERA meta-analysis (combining non-Hispanic whites and Hispanic/Latinos) at Bonferroni significance ($P < 0.05/30 = 0.00167$), including 1 SNP at *BNC2* and 5 at *OCA2/HERC2*, and among those, 11 reached genome-wide level of significance: 2 at *SLC45A2*, 1 at *IRF4*, 2 at *TYR*, 4 at 16q24.3 (*DEF8-MC1R*), and 2 at *RALY* (Supplementary Data 1), consistent with previous GWAS of cSCC[18,19]. Three additional SNPs (i.e., 1 at *TYRP1* and 2 at 16q24.3) reached nominal significance ($P < 0.05$).

**Genetic ancestry and known cSCC-associated loci**. To determine whether known cSCC-associated loci explain the remaining genetic ancestry associations (after considering genetically determined skin pigmentation), we repeated the ancestry analysis, including a genetic risk score (GRS) based on 14 reported cSCC-SNPs (as two other reported SNPs, rs192481803 and rs74899442, were not polymorphic (MAF < 1.0%) in the GERA Hispanic/Latino and non-Hispanic white subjects, and so were excluded from the GRS and subsequent analyses) (Supplementary Table 4 and Methods online). The 14 SNPs were confirmed to be associated with cSCC risk in GERA at Bonferroni significance ($P < 0.05/14 = 0.0036$) (Supplementary Data 2). When including the GRS in addition to the HIrisPlex predicted skin pigments, the ancestry associations were slightly attenuated in Hispanic/Latinos (Model 3: $\beta = 56.54$, $P = 8.54 \times 10^{-7}$ for PC1 and $\beta = 28.44$, $P = 0.069$ for PC2) but were not attenuated further in non-Hispanic whites (Model 3: $\beta = 17.20$, $P = 1.07 \times 10^{-20}$ for PC1 and $\beta = -18.33$, $P = 4.41 \times 10^{-26}$ for PC2) (Table 2 and Supplementary Tables 1 and 2). Thus, the genetic ancestry effects in Hispanic/Latinos and non-Hispanic whites cannot be explained completely by current cSCC-SNPs.

**Shared genetic risk loci for cSCC and skin pigmentation traits**. Of the 14 reported cSCC-SNPs and the 36 known skin pigmentation genetic variants, 5 SNPs were overlapping: *IRF4* rs12203592, *TYR* rs1126809, *OCA2* rs1800407, *MC1R* rs1805007, and *RALY* rs6059655. Further, several SNPs known to influence skin pigmentation or cSCC risk were located within the same locus and were in strong linkage disequilibrium (LD), including rs28777 and rs35407 at *SLC45A2* ($R^2 = 0.98$, $D' = 1$), rs12913832 and rs12916300 at *HERC2* ($R^2 = 0.84$, $D' = 0.96$), and rs8051733 and rs4268748 at *DEF8* ($R^2 = 0.73$, $D' = 0.97$) (Supplementary Table 5). This suggests that those SNPs represent the same signal at those loci and that those SNPs are most likely proxies of causal variants influencing both skin pigmentation and cSCC risk. It is noteworthy that rs683 at *TYRP1*, known to influence skin color pigment but not previously associated with cSCC risk, showed suggestive association with cSCC risk in our GERA non-Hispanic white sample ($P = 0.0064$) (Supplementary Data 1). *TYRP1* encodes the melanosomal enzyme tyrosinase-related protein 1, and mutations in this gene can cause albinism and confer increased risk for familial cutaneous melanoma[27–29].

Of the 14 cSCC-SNPs, five (35.7%) were not associated with HIrisPlex-S skin pigment probabilities in GERA (non-Hispanic whites or Hispanic/Latinos) ($P \geq 0.05$), including SNPs at *FOXP1*, *TPRG1/TP63*, *HLA-DQA1*, *AHR*, and *SEC16A* loci (Supplementary Data 3). Of the five cSCC-SNPs that were not associated with HIrisPlex-S skin pigment probabilities, four (other than *SEC16A* rs57994353) were strongly associated with genetic ancestry PC1 or PC2 (Supplementary Table 6). That is, genetic risk factors other than those that affect pigment are also responsible for the association of ancestry and cSCC risk.

**Independent signals for cSCC at skin pigmentation loci**. To determine whether there were additional independent signals at the skin pigmentation loci for cSCC risk, we conducted conditional analyses of the skin pigmentation loci for cSCC association by including all the 14 known cSCC-SNPs in the regression model. We found independent signals at four known skin pigmentation loci among non-Hispanic whites (i.e., *SLC45A2*, *HERC2*, *MC1R*, and *DEF8*) (Supplementary Data 4). Most of these SNPs have been previously reported to be associated with pigmentation traits[30–32], including rs16891982 at *SLC45A2*, rs1667394 at *OCA2/HERC2*, and rs1805008 at *MC1R*.

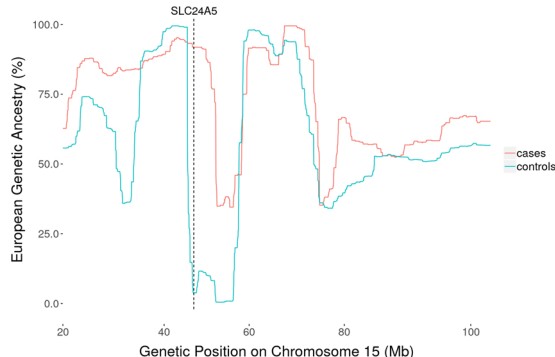

**Fig. 2 Association of local ancestry at *SLC24A5* and cSCC risk in GERA Hispanic/Latinos.** A local ancestry analyses was conducted in 8490 GERA Hispanic/Latinos (294 cSCC cases and 8196 controls). A significant association between cSCC risk and local European genetic ancestry was detected at *SLC24A5*, where on average cases were of 91.8% European descent compared with controls at 3.8% ($P = 1.71 \times 10^{-4}$).

**Local ancestry**. To determine whether each of the 16 known cSCC- and/or skin pigmentation-associated loci show evidence of increased European ancestry in cSCC cases compared to controls, we conducted local ancestry analyses in GERA Hispanic/Latinos using a method for local ancestry inference, which leverages the structure of LD in the ancestral population (LAMP-LD)[33]. After correction for testing multiple loci, we detected a significant association between cSCC risk and local European genetic ancestry at *SLC24A5*, where on average cases were of 91.8% European descent compared with controls at 3.8% (*SLC24A5* $P = 1.71 \times 10^{-4}$) (Supplementary Table 7 and Fig. 2).

Interestingly, we identified an association between cSCC and the well-known skin pigment variant *Ala111Thr* (rs1426654) in the *SLC24A5* gene in Hispanic/Latinos, a population in which it is polymorphic (Supplementary Data 1). This association of *SLC24A5* skin pigment-associated genetic variant with cSCC in Hispanic/Latinos is notable, as not previously reported, and could explain the local ancestry effect.

**SNP prioritization and annotations**. To identify which genetic variants within the 18 cSCC and/or skin pigmentation loci (14 cSCC + 4 conditional hits at known skin pigmentation loci) are likely to be causal, we computed each variant's ability to explain the observed signal and derived the smallest set of variants that included the causal variant with 95% probability[34]. In each of the 18 loci, the corresponding 18 credible sets contained from 1 to 3868 variants in Hispanic/Latinos (7391 total variants, Supplementary Data 5) and from 1 to 1104 variants in non-Hispanic whites (1373 total variants, Supplementary Data 6). Interestingly, two variants (i.e., *IRF4* rs12203592 and *SLC45A2* rs16891982) had >50% probability of being causal in both Hispanic/Latinos and non-Hispanic whites. Additional five variants (i.e. *TYR* rs1126809, *RALY* rs6059655, *BNC2-CNTLN* rs74664507, *DEF8* rs4268748, and *HERC2* rs12916300) had >50% probability of being causal in non-Hispanic whites.

**Genetic ancestry associations and sun exposure**. To determine whether chronic UVR/sun exposure mediates the observed genetic ancestry associations, we repeated the ancestry analysis, including AK as a covariate. In GERA, among cSCC cases, 65.5% had AK (Table 1). When including AK in addition to the HIrisPlex predicted skin pigments and the GRS, the ancestry associations were attenuated further but not eliminated (Model 4: $\beta = 41.30$, $P = 5.67 \times 10^{-4}$ for PC1 and $\beta = 18.67$, $P = 0.22$ for PC2 in Hispanic/Latinos; $\beta = 11.49$, $P = 1.52 \times 10^{-9}$ for PC1 and

$\beta = -12.99$, $P = 5.37 \times 10^{-13}$ for PC2 in non-Hispanic whites) (Table 2 and Supplementary Tables 1 and 2).

As a secondary analysis, we assessed genetic ancestry associations with cSCC arising in anatomical sites exposed to the sun compared to sun-protected anatomical sites. Among cSCC cases, 87.7% had tumors at sun-exposed sites only compared to 3.7% for sun-protected only, and 4.1% for both (Table 1). While we found significant associations between genetic ancestry and cSCC risk at anatomical sites exposed to the sun ($\beta = 127.68$, $P = 2.11 \times 10^{-39}$ for PC1 and $\beta = 72.29$, $P = 8.69 \times 10^{-4}$ for PC2 in Hispanic/Latinos; $\beta = 30.62$, $P = 1.16 \times 10^{-60}$ for PC1 and $\beta = -25.82$, $P = 4.83 \times 10^{-46}$ for PC2 in non-Hispanic whites), the effects of these associations were reduced but not eliminated at anatomical sites unexposed to the sun (Model 1: $\beta = 47.93$, $P = 0.094$ for PC1 and $\beta = -5.27$, $P = 0.78$ for PC2 in Hispanic/Latinos; $\beta = 20.37$, $P = 0.0094$ for PC1 and $\beta = -17.07$, $P = 0.024$ for PC2 in non-Hispanic whites) (Supplementary Fig. 3 and Supplementary Table 8). After adjusting for skin pigment traits, the genetic ancestry associations with cSCC at anatomical sites unexposed to the sun were attenuated further and no longer significant (Model 2: $\beta = 37.84$, $P = 0.32$ for PC1 and $\beta = -7.16$, $P = 0.71$ for PC2 in Hispanic/Latinos; $\beta = 9.83$, $P = 0.23$ for PC1 and $\beta = -11.85$, $P = 0.12$ for PC2 in non-Hispanic whites).

**Variance explained by each cSCC risk factor**. Of interest, when examining the amount of variance explained by each covariate (i.e., genetic ancestry PCs, HIrisPlex-S skin pigment probabilities, GRS based on 14 known cSCC-associated SNPs, or history of AK), we noted that the genetic ancestry effect was much stronger in the Hispanic/Latinos ($R^2 = 12.69\%$) than in the non-Hispanic whites ($R^2 = 1.23\%$) (Table 3). This is likely due to the greater contrast between Native American/African genetic background versus European compared to the gradient of genetic ancestry in Europe. We also note that the GRS explained more variance in the Hispanic/Latinos ($R^2 = 1.01\%$) versus non-Hispanic whites ($R^2 = 0.43\%$), as also did the skin-pigmentation SNPs ($R^2 = 3.83\%$ versus 1.52%). We also conducted formal model comparison analyses (i.e., likelihood-ratio tests) that demonstrate that the more complex model (Model 4) is warranted over the other models (Table 4).

**Table 3 Variances explained by each cSCC risk factor.**

| cSCC risk factor | Model | Hispanic/Latinos | non-Hispanic whites |
|---|---|---|---|
| | | Proportion (%) of variance explained adjusted $R^2$ | |
| Genetic ancestry PCs | 1 | 12.69 | 1.23 |
| HIrisPlex-S skin pigmentation predictions | 2 | 3.83 | 1.52 |
| GRS based on 14 known cSCC-SNPs | 3 | 1.01 | 0.43 |
| Sun exposure (history of AK) | 4 | 6.68 | 6.40 |

PC principal components, GRS genetic risk score, AK actinic keratosis.

**Table 4 Model comparison analyses using likelihood-ratio tests.**

| | Model 1 versus 2 | Model 2 versus 3 | Model 3 versus 4 |
|---|---|---|---|
| Added variables | HirisPlex-S skin pigmentation probabilities | GRS (based on 14 known cSCC variants) | Sun exposure (history of AK) |
| GERA ethnic group | Likelihood-ratio test, P value | | |
| Non-Hispanic white | $2.96 \times 10^{-188}$ | $5.73 \times 10^{-54}$ | $<2.23 \times 10^{-308}$ |
| Hispanic/Latino | $5.58 \times 10^{-17}$ | $1.35 \times 10^{-5}$ | $7.46 \times 10^{-29}$ |

**Genetic ancestry and ncSCC risk**. Finally, we assessed whether genetic ancestry had a similar effect on ncSCC compared to cSCC. ncSCC arise in other area epithelial-lined surfaces of the body, including the oral cavity, esophagus, lung, and cervix, and development is typically not related to sun exposure. We investigated whether the first two PCs were associated with ncSCC risk within GERA Hispanic/Latinos and non-Hispanic whites. We identified 1065 ncSCC cases and 75,601 controls, and the prevalence of ncSCC was on average 1.4% and was very similar across ethnic groups (Supplementary Table 9). None of the PCs was significantly associated with ncSCC risk in Hispanic/Latinos or non-Hispanic whites (Supplementary Fig. 4 and Supplementary Table 10). We also investigated whether the 36 skin pigmentation genetic variants and 14 cSCC-associated variants were associated with ncSCC risk in GERA (Supplementary Data 7 and 8). While we observed an association at Bonferroni-level significance in GERA Hispanic/Latinos between the pigment-associated *MC1R* rs885479-G (odds ratio (OR) = 0.59, $P = 3.78 \times 10^{-4}$) and ncSCC, none of the 14 previously reported cSCC-associated variants was associated with ncSCC risk. Further, we did not observe significant associations with ncSCC risk and any of these variants in non-Hispanic whites. Thus, genetic ancestry and genetically determined skin pigmentation appear to be exclusive characteristics of cutaneous SCC and not ncSCC.

**Discussion**

In this large multiethnic study, we found that the risk of cSCC was associated with genetic ancestry within Hispanic/Latinos and non-Hispanic whites. Specifically, a greater risk of cSCC was associated with higher European (compared to Native American or African) ancestry among Hispanic/Latinos and northern and western (compared to southern and eastern) European ancestry among non-Hispanic whites. After considering genetic variants known to influence skin pigmentation or cSCC risk and considering AK, which is a marker of chronic sun exposure, these genetic ancestry associations were significantly attenuated but not eliminated. Moreover, the associations between genetic ancestry and cSCC risk were attenuated but not eliminated in cases where the cSCC arose in sun-protected anatomical sites compared to the sun-exposed anatomical sites. Finally, we did not observe significant associations between genetic ancestry and ncSCC risk. These findings suggest that genetically determined skin pigmentation can explain much of the ancestry effects in both Hispanic/Latinos and non-Hispanic whites and that additional loci associated with genetic ancestry remain to be discovered. However, we found evidence for an ancestry effect on the non-pigment genetic loci associated with the risk of cSCC. This suggests that these additional cSCC loci that remain to be discovered may or may not be associated with skin pigmentation and the genetic ancestry effect may extend beyond just skin pigmentation, per se.

This is, to our knowledge, the first study to examine genetic risk factors for cSCC in Hispanic/Latino populations, and we identified significant associations between the risk of cSCC and genetic ancestry, genetic predictors of skin pigmentation, sun exposure, and cSCC risk loci previously identified in populations of European descent. We also found evidence that the percentage of European ancestry at *SLC24A5* was strongly correlated with cSCC case status ($P = 1.71 \times 10^{-4}$) in Hispanic/Latinos. cSCC cases contained on average 91.8% European ancestral alleles at the *SLC24A5* locus compared to 3.8% in controls, suggesting that European-specific variation may contribute to cSCC risk at this locus. *SLC24A5* is a well-known target of positive selection, associated with lighter skin color in admixed populations[35,36]. Indeed, we showed that the well-known amino acid substitution at this locus, *Ala111Thr*, previously shown to be continentally

dichotomous between Europeans and the rest of the world, and strongly associated with pigment in populations in which it is polymorphic (e.g., Hispanics and South Asians)[37–39], is also a risk factor for cSCC in our Hispanic/Latino subjects. This is an important novel cSCC association uniquely found in this population, to date. Future studies can investigate whether this variant may also be an important risk factor in African-American populations.

Among non-Hispanic whites, we observed that a higher cSCC risk is associated with northern European genetic ancestry as compared to southern European ancestry; however, we also found a greater risk of cSCC in subjects with northwestern European ancestry compared to northeastern European ancestry. Observational studies have reported a north–south gradient of cSCC susceptibility in European countries with cSCC incidence rate rising with increasing latitude[10]. This latitudinal gradient, which is paradoxically inversely correlated with ultraviolet light indices, a known risk factor for cSCC, is thought to be a result of the gradient of skin color observed in Europe, as the proportion of individuals with darker skin types increase with decreasing latitude[10]. After considering the effect of skin pigmentation, the ancestry associations were significantly attenuated but not eliminated. This suggests that skin pigmentation is an important contributor to cSCC risk and the European latitude gradient, but does not fully explain it.

We recognize several potential limitations of our study. First, any association with genetic ancestry may represent not only variation in genetic risk factors but also other factors that correlate with genetic ancestry, such as environmental or sociocultural factors[40,41], and so it is possible that the residual ancestry associations that we observed may reflect non-genetic risk factors for cSCC. Second, the skin pigmentation traits that we used in the current study were based on HIrisPlex-S probabilities estimated using genetic information, as direct pigment measures are not available in the GERA cohort. This may result in underestimates of the effects of pigment variation due to phenotype misclassification. However, HIrisPlex-S system has been shown to predict pigment traits accurately when using all the required SNPs (36 for skin color)[24], with average prediction accuracy of 91% for skin color. Third, we used a clinical surrogate for cumulative lifetime exposure to UVR, namely clinician-diagnosed AKs, which could be a possible intermediate step in the development of cSCC. We also recognize that specifying tumor locations according to sun exposure may depend on individual behavior, which could affect the generalizability of our sun-exposed versus sun-protected findings. However, our findings from ncSCC, which are consistent with these results, are unlikely to be affected by either of these limitations. Finally, certain analytical strata of our study have relatively few cases (e.g., low number of Hispanic/Latino cases compared to non-Hispanic whites, and low number of sun-protected cases) that may limit our power to detect significant associations in these cases.

Our study also has important strengths. It is based on a unique and very large multiethnic cohort of individuals, who were all members of a single integrated health delivery system. Further, this is the first study that enabled cross populations comparison (i.e., between Hispanic/Latino and non-Hispanic whites) by taking into account genetic ancestry and known risk factors for cSCC.

In conclusion, we reported variation in cSCC risk not only across populations, but also within Hispanic/Latino and non-Hispanic white populations. This variation in cSCC risk is due in part to genetic variation in skin pigmentation genes and its interaction with sun exposure leading to higher risk. These results, based on additive models, underscore the importance of incorporating genetic ancestry, skin pigmentation traits, and sun

exposure into genetic analyses of skin cancer. However, additional cSCC genetic risk loci and other cSCC risk factors that correlate with genetic ancestry remain to be discovered. Thus, our study highlights the importance of considering genetic structure when mapping the genetic basis of a disease phenotype showing correlation a between prevalence and geographic origin. Our findings identifying genetic ancestry as an important risk factor for cSCC enable identification of the most vulnerable members of a population at high risk for cSCC and can lead to prevention strategies for this highly prevalent cancer[42–44].

## Methods

### Study population
*Cutaneous squamous cell carcinoma.* Potential eligible study subjects were adult GERA cohort participants who had no diagnostic codes for rare genetic disorders associated with increased cSCC risk[18]. cSCC cases were defined as subjects whose pathology records were consistent with incident cSCC (invasive or in situ, excluding anogenital and mucosal SCCs). Controls were subjects with no pathology records consistent with cSCC. After quality control (QC) was applied to all potential eligible subjects, a total of 97,582 subjects were eligible for analysis, consisting of 11,396 cSCC cases and 86,186 controls from four ethnic groups (80.6% non-Hispanic white, 8.7% Hispanic/Latino, 7.5% East Asian, and 3.2% African American).

*Non-cutaneous squamous cell carcinoma.* The KPNC (Kaiser Permanente Northern California) Cancer Registry captured all SCC cases in the GERA cohort. We identified individuals as ncSCC cases by excluding individuals with cSCC from the overall SCC case group. Controls were GERA participants who were neither cSCC nor ncSCC cases.

*History of AK.* AK is a clinical surrogate of cumulative lifetime exposure to UVR. In a large, population-based study of skin cancer, the presence of AKs correlated most strongly with the variables estimating thhe cumulative number of hours spent outdoors in a self-administered questionnaire[45].

All study procedures were approved by the Institutional Review Board of the KPNC Institutional Review Board. Written informed consent was obtained from all participants.

### Anatomic site of tumor
Anatomic locations of the cSCC were classified according to the potential of the anatomic area to be chronically exposed to the sun, rather than more likely to be covered by clothing, as follows: sun-exposed (head, neck, hands, feets, and legs) and sun-protected (abdomen, buttocks, and hips).

### Genotyping and imputation and QC
DNA samples were genotyped on Affymetrix Axiom arrays as previously described[46,47]. Briefly, SNPs with allele frequency difference ≤0.15 between males and females for autosomal markers, and genotype concordance rate >0.75 across duplicate samples were included[47]. Approximately 94% of samples and >98% of genetic markers assayed passed QC procedures. SNPs with genotype call rates <90% and SNPs with a MAF <1% were excluded. Following genotyping QC, we conducted statistical imputation of additional genetic variants. Following the pre-phasing of genotypes with Shape-IT v2. r7271958, variants were imputed from the cosmopolitan 1000 Genomes Project reference panel (phase I-integrated release; http://1000genomes.org) using IMPUTE2 v2.3.0. As a QC metric, we used the info $r^2$ from IMPUTE2, which is an estimate of the correlation of the imputed genotype to the true genotype. Variants with an imputation $r^2 < 0.3$ were excluded, and we restricted the imputation to SNPs that had a minor allele count ≥20 in the reference panel.

### Defining genetic ancestry
The genetic structure of GERA participants has been already characterized using genome-wide genotypes and PCs analysis[21] and identified 10 and 6 ancestry PCs reflecting genetic ancestry among non-Hispanic whites, and the other ethnic groups, respectively[21]. In this study[21], genetic ancestry PCs were calculated by Eigenstrat[48] v4.2. In the current study, for genetic ancestry analyses, for each ethnic group, we focused on PC1 and PC2 because they are the only PCSs that are geographically interpretable. In non-Hispanic whites, PC1 represents a European northwestern versus southeastern cline, and PC2 represents a European northeastern versus southwestern cline. In Hispanic/Latinos, PC1 represents greater European versus Native American ancestry and PC2 represents greater African versus European ancestry.

### Plots of cSCC prevalence versus genetic ancestry
To visualize the cSCC prevalence distribution by the ancestry PCs, we created an ancestry contour figure based on a smoothed distribution of each individual $i$'s cSCC phenotype (affected versus not) using a radial kernel density estimate weighted on the distance to each other $j$th individual, $\sum j\phi(\{d(i,j)/\max_{i',j'}[d(i', j')] \times k)\})$, where $\phi(.)$ is the standard normal density distribution, $k$ is the smoothing value (15 for non-Hispanic white

and Hispanic/Latino ethnic groups), and d(i,j) is the Euclidean distance based on the first two PCs. A more detailed description of the method used to generate these plots has been published[49]. Nationality subgroup labels were derived from the Human Genome Diversity Project for the visual representation of different subgroups from that sample[21]. We have previously used these contour graphs to visualize genetic ancestry in other disorders in the GERA cohort[40,50]. Ancestry contour figures using the viridis colors are also provided in the Supplementary Information.

**HIrisPlex-S program and predicted skin pigments**. To determine whether the genetic ancestry associations with cSCC were due to differences in skin pigmentation, we predicted skin pigment traits (i.e., "dark skin," "dark to black skin," "intermediate skin," "very pale skin," and "pale skin") using HIrisPlex-S[24–26], a program that utilizes known pigment genetic variants to generate probabilities of different pigment traits. The HIrisPlex-S predictive model includes 36 SNPs for skin color (Supplementary Table 3). We did consider using a five-category level (very pale, pale, intermediate, dark, and dark-black) of the HIrisPlex-S skin pigmentation predictions and examined the distribution of all the categories in GERA non-Hispanic whites (Fig. 3). Because the distribution of the categories "very pale" and "dark to black" were not well differentiated from "pale" and "dark," respectively, we decided to combine "dark" and "dark to black skin" into a single category, and "very pale" and "pale" into a single category. Thus, by focusing on a three-category level (very pale + pale, intermediate, dark + dark-black), we improved the predictive values for skin pigmentation compared to the original five-category level. To determine whether predicted skin pigmentation traits explain the observed associations of genetic ancestry with cSCC risk, we included the probabilities for "dark + dark to black," and intermediate skin colors in the logistic regression models. Because the skin pigment probabilities sum to 1, we had to exclude at least one skin pigment probability. We chose to exclude "very pale + pale skin" from our models. As a result, the direction of the effect estimates of the skin pigment probabilities represents darker pigment.

**GRS based on cSCC-associated SNPs**. To determine whether the known cSCC-associated loci could explain the observed associations of genetic ancestry with the risk of cSCC, we also included a weighted GRS in the logistic regression model. For SNP selection, we first investigated whether 16 SNPs associated with cSCC risk at a genome-wide significance level in previous GWAS studies[18,19,51] were also associated with cSCC in GERA (Supplementary Data 2). SNPs rs192481803 and rs74899442 were not polymorphic (MAF < 1.0%) in any of the four GERA ethnic groups, and so were excluded from the GRS and subsequent analyses. The 14 remaining SNPs (Supplementary Table 4) were confirmed to be associated with cSCC risk in GERA at Bonferroni significance ($P < 0.05/14 = 0.0036$) (Supplementary Data 2). The GRS was built on these 14 cSCC-SNPs by summing up the additive coding of each SNP weighted by the effect size ascertained from the original studies. As all previous GWAS of cSCC have been conducted in cohorts of European ancestry, we also generated unweighted GRS for the analyses conducted in the Hispanic/Latino sample. Results were similar using unweighted or weighted GRS in Hispanic/Latinos (Supplementary Table 11). Association of genetic ancestry with each skin pigmentation trait- and cSCC-associated SNP are presented in Supplementary Table 6.

**Local ancestry**. To determine whether the proportion of European ancestry at the 16 known cSCC- and/or skin pigmentation-associated loci could account for cSCC risk in Hispanic-Latinos, we conducted local ancestry analyses using LAMP-LD[33].

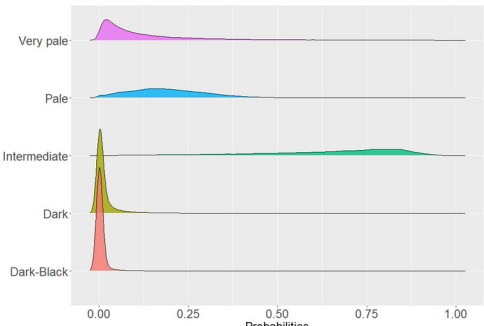

**Fig. 3 Distribution of the five-category level of the HIrisPlex-S skin pigmentation predictions in GERA non-Hispanic whites.** Skin pigment traits (i.e., "dark skin," "dark to black skin," "intermediate skin," "very pale skin," and "pale skin") were predicted using HIrisPlex-S program based on 36 SNPs known to influence skin color. We did consider using a five-category level (very pale, pale, intermediate, dark, and dark-black) of the HIrisPlex-S skin pigmentation predictions and examined the distribution of all the categories in GERA non-Hispanic whites.

We used the "HGDP-CEPH Human Genome Diversity Cell Line Panel"[52] (https://hagsc.org/hgdp/). Association testing between cSCC status and percent European ancestry at each locus was performed using the nonparametric test statistic proposed by Montana and Pritchard for admixture mapping[53]. For this analysis, a Bonferroni-corrected significance threshold of $0.05/16 = 3.1 \times 10^{-3}$ (accounting for the number of loci tested) was applied. We used the R package "ggplot2" to generate the local ancestry at SLC24A5 plot presented in Fig. 2.

**Variants prioritization**. To identify which genetic variants within the 18 cSCC and/or skin pigmentation (14 cSCC + 4 conditional hits at known skin pigmentation loci) are likely to be causal, we used a Bayesian approach (CAVIARBF)[34]. For each of the 18 loci, we computed each variant's capacity to explain the identified signal within a 2 Mb window (±1.0 Mb with respect to the original top variant) and derived the smallest set of variants that included the causal variant with 95% probability (95% credible set). A total of 7391 variants within 265 annotated genes were included in these 18 credible sets for the Hispanic/Latino analysis (Supplementary Data 5) and 1374 variants within 45 annotated genes were included in these 18 credible sets for the non-Hispanic white analysis (Supplementary Data 6).

**Statistical analysis**. We used a logistic regression model to examine the impact of ancestry on cSCC risk using R version 3.4.1 with the following covariates: age, sex, and ancestry PCs (Model 1). For all the analyses, we included as covariates the top 10 ancestry PCs for the non-Hispanic whites, whereas we included the top 6 ancestry PCs for the Hispanic/Latinos. To adjust for genetic ancestry, we also included the percentage of Ashkenazi (ASHK) Jewish ancestry as a covariate for the non-Hispanic white ethnic group analysis. We present the full multivariate logistic regression models of cSCC and ncSCC in Supplementary Tables 1, 2, 6, and 8, including all the PCs for each ethnic group, and adjusting for age and sex. In Model 2, in addition to all covariates included in Model 1, we added skin pigment probabilities from HIrisPlex-S. In Model 3, in addition to all covariates included in Model 2, we added the GRS described above. In Model 4, in addition to all covariates included in Model 3, we added history of AK as a covariate, reflecting sun exposure. The proportion of variance explained by each covariate (i.e., genetic ancestry PCs, HIrisPlex-S skin pigment probabilities, GRS based on 14 known cSCC-associated SNPs, or history of AK) and for each ethnic group (i.e., non-Hispanic white or Hispanic/Latino) are reported in Table 3. For the conditional analysis of the skin pigmentation loci for cSCC association, we incorporated 14 previously reported cSCC risk SNPs as covariates.

**Analyses of ncSCC**. To assess whether ncSCC followed a similar pattern of genetic ancestry associations to cSCC, we conducted a parallel analysis of ncSCC in GERA, evaluating the association of genetic ancestry and potential modification by skin pigment and previously reported cSCC-SNPs.

**Reporting summary**. Further information on research design is available in the Nature Research Reporting Summary linked to this article.

## Data availability
Genotype data of GERA participants are available from the database of Genotypes and Phenotypes (dbGaP) under accession phs000674.v2.p2. This includes individuals who consented to having their data shared with dbGaP. The complete GERA data are available upon application to the KP Research Bank (https://researchbank.kaiserpermanente.org/).

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

## Acknowledgements

This research was funded by a grant from the National Cancer Institute (R01CA166672 and R01CA231264 to M.M.A.), the National Institute of Arthritis and Musculoskeletal and Skin Diseases (K24AR069760 to M.M.A.), and the National Institute on Aging, National Institute of Mental Health, and National Institute of Health Common Fund (RC2 AG036607 to N.R. and C.S.). E.J. and H.C. were supported by grants from the National Eye Institute (R01 EY027004), the National Institute of Diabetes and Digestive and Kidney Diseases (R01 DK116738), and from the National Cancer Institute (R01 CA2416323).

## Author contributions

E.J., H.C., and M.M.A. conceived and designed the study. T.J.H., M.N.K, N.R., C.S., and E.J. were involved in the genotyping and quality control. Y.B. performed the ancestry principal component analyses. T.J.H. performed the imputation analyses. J.Y. extracted phenotype data and performed statistical analyses. E.J., H.C., N.R., and M.M.A. interpreted the results of analyses and contributed to the drafting and critical review of the manuscript.

## Competing interests

M.M.A. has received research funding from Valeant Pharmaceuticals on an unrelated topic. The remaining authors declare no competing financial or non-financial interests. H.C. is an Editorial Board Member for *Communications Biology*, but was not involved in the editorial review of, nor the decision to publish this article.
