## [Peer Review File · Communications Biology]

Reviewers' comments:

Reviewer #1 (Remarks to the Author):

Jorgenson et. al explored the associations between genetic ancestry, genetically predicted skin pigmentation, sun exposure, and risk of cutaneous squamous cell carcinoma. It is a well-executed study. The conclusion is original especially among individuals of Hispanic ancestry.

The statistical process is appropriate, and all the conclusions are valid. One issue came into my mind is that the variance explained by these respective "risk" factors for cutaneous squamous cell carcinoma. This information could be provided in supplementary tables.

A minor error at page 9. I could not find this information at supplementary table 6.

"After adjusting for skin pigment traits, the genetic ancestry associations with cSCC at anatomical sites unexposed to the sun were attenuated further and no longer significant (Model 2: $\beta=58.62$, $P=0.19$ for PC1 and $\beta=-4.13$, $P=0.83$ for PC2 in Hispanic/Latinos; $\beta=10.51$, $P=0.21$ for PC1 and $\beta=-10.96$, $P = 0.16$ for PC2 in non-Hispanic whites)."

Reviewer #2 (Remarks to the Author):

Jorgenson and colleagues report a positive association between European ancestry (in Hispanics) and northern European ancestry (non-Hispanic whites) and risk of cSCC after adjustment for genetically predicted skin pigmentation as well as known cSCC loci and actinic keratosis. This effect was significant only for lesions in sun-exposed regions in Hispanics. These results were not replicated for ncSCC. The methods are sound, and the manuscript is very well written. Overall, it makes a significant contribution to the literature and demonstrates the existence of additional risk variants for these tumors that are not related to skin pigmentation, One potential limitation of this study is the relatively low number of Hispanic and sun-protected cases, which may be why some associations within this group were non-significant after adjustment as compared to non-Hispanic white cases. I understand that these cSCC cases are much less common, so I am not sure this necessarily warrants discussion in the manuscript.

Reviewer #3 (Remarks to the Author):

Summary:

In a novel study, Jorgenson et al examine the roles of genetic ancestry, skin pigmentation, and sun exposure for cSCC risk in Hispanics/Latinos and non-Hispanic Whites in GERA cohort. Using principal components computed from genotyping data as a proxy for ancestry, they found significant associations between the two leading principal components and cSCC. The authors then demonstrate that the estimated effects of ancestry are attenuated after conditioning on genetically predicted skin pigmentation, known cSCC risk loci, and a proxy variable for sun exposure. They also performed stratified analysis by anatomical sites of cSCC and a similar analysis in ncSCC to demonstrate the heterogeneous effect of genetic ancestries on different cSCC types. The results were clearly presented and organized in way that made it straightforward to read and interpret. With that said, I have a few comments the authors should address.

Major Comments:

1. Several main analyses rely on the use of predicted skin pigmentation from a specific model. I certainly appreciate the fact that the tool has been successfully published and described previously, it would lend much more impact to the current paper if it is at all possible to perform some validation of the predicted pigmentation in the GERA cohort.

2. Throughout the article the authors successively build upon models to demonstrate the weakened effect of ancestry on cSCC risk. While some models make clear improvements, others are more uncertain. I appreciate the inclusion of p-values for the effect sizes at each stage, however, the manuscript would be significantly strengthened with a formal model comparison analysis. That is, as model complexity increases, can the authors demonstrate that the more complex model is warranted with a likelihood ratio test (or equivalent procedure)? The stronger the evidence under this procedure, the more likely that the component of risk explained by global ancestry is not fully mediated by the additional variables.

3. The authors report cSCC GWAS results conditional on 14 known risk SNPs. These 14 known SNPs were identified in non-Hispanic whites, and it isn't clear whether the resulting conditional findings claimed to be independent associations are truly independent, or just the result of impartial tagging of the underlying causal variants. That is, if the "known markers" are tagging causal variants, they will be less effective tag SNPs in a Hispanic population with different linkage disequilibrium patterns. Then, a conditional analysis will result in false positive "independent" hits.

To remedy this, I suggest the authors simply perform a statistical fine-mapping at the risk regions in their own data, to identify what likely is causal. This will more clearly reflect what variants are contributing to risk.

4. There is currently not enough information on the models used to test for association between SNP and cSCC risk. This is crucial because it is not clear whether population structure was sufficiently controlled.

5. It would be interesting to see the effect of local ancestry at both the known cSCC sites and pigmentation sites in the GERA cohort. That is, for the Hispanic individuals, can the authors call local ancestries (using RFMix or similar tool) and perform a localized admixture scan? These results would help bolster the currently presented findings.

Minor Comments:

1. The gradient figures are very nice and incredibly helpful to see the geographic/ancestral distribution of variables. It would be helpful to use a more colorblind-friendly palette (e.g., viridis).

2. The authors concluded (p.12) the variation in cSCC risk within Hispanic/Latino and non-Hispanic white populations "is due in part to skin pigmentation genes and the interaction of those genes with sun exposure leading to higher risk." I would consider rewording this statement as the only models investigated were additive with respect to risk. An interesting analysis would be an interaction between predicted pigmentation and sun exposure.

3. Skin pigmentation were originally classified by the HIrisPlex-S into 5 levels. However, the authors could explain why they regrouped the pigmentation into 3 levels: "dark skin + dark to black skin", "intermediate skin", "very pale skin + pale skin". I wonder how the 5 levels will be correlated with PCs, whether there will be a linear trend in their effects on cSCC, and how much associations between cSCC and PCs would be left when the original scale was used.

Reviewers' comments:

Reviewer #1 (Remarks to the Author):

Jorgenson et. al explored the associations between genetic ancestry, genetically predicted skin pigmentation, sun exposure, and risk of cutaneous squamous cell carcinoma. It is a well-executed study. The conclusion is original especially among individuals of Hispanic ancestry.

We thank the reviewer for the positive feedback.

The statistical process is appropriate, and all the conclusions are valid. One issue came into my mind is that the variance explained by these respective “risk” factors for cutaneous squamous cell carcinoma. This information could be provided in supplementary tables.

We have added some text in the Methods section (as follows) and we also provided the proportion of variance explained by each cSCC risk factor in a new Supplemental Table (Supplementary Table 12):

“The proportion of variance explained by each covariate (i.e. genetic ancestry PCs, HIrisPlex-S skin pigment probabilities, GRS based on 14 known cSCC-associated SNPs, or history of AK) and for each ethnic group (i.e. non-Hispanic white or Hispanic/Latino) are reported in Supplementary Table 12.”

cSCC risk factor	Model	Hispanic/Latinos	non-Hispanic whites
		Proportion (%) of Variance Explained Adjusted R ²	
Genetic ancestry PCs	1	12.69	1.23
HIrisPlex-S skin pigmentation predictions	2	3.83	1.52
GRS (based on 14 known cSCC-associated SNPs)	3	1.01	0.43
Sun exposure (History of actinic keratosis)	4	6.68	6.40

A minor error at page 9. I could not find this information at supplementary table 6.

“After adjusting for skin pigment traits, the genetic ancestry associations with cSCC at anatomical sites unexposed to the sun were attenuated further and no longer significant (Model 2: $\beta=58.62$, $P=0.19$ for PC1 and $\beta=-4.13$, $P=0.83$ for PC2 in Hispanic/Latinos; $\beta=10.51$, $P=0.21$ for PC1 and $\beta=-10.96$, $P = 0.16$ for PC2 in non-Hispanic whites).”

We thank the reviewer for pointing out this error in the manuscript. We have revised this text to provide results that are presented in Supplementary Table 6, as follows:

“After adjusting for skin pigment traits, the genetic ancestry associations with cSCC at anatomical sites unexposed to the sun were attenuated further and no longer significant (Model 2: $\beta=37.84$, $P=0.32$ for PC1 and $\beta=-7.16$, $P=0.71$ for PC2 in Hispanic/Latinos; $\beta=9.83$, $P=0.23$ for PC1 and $\beta=-11.85$, $P=0.12$ for PC2 in non-Hispanic whites).”

Reviewer #2 (Remarks to the Author):

Jorgenson and colleagues report a positive association between European ancestry (in Hispanics) and northern European ancestry (non-Hispanic whites) and risk of cSCC after adjustment for genetically predicted skin pigmentation as well as known cSCC loci and actinic keratosis. This effect was significant only for lesions in sun-exposed regions in Hispanics. These results were not replicated for ncSCC. The methods are sound, and the manuscript is very well written. Overall, it makes a significant contribution to the literature and demonstrates the existence of additional risk variants for these tumors that are not

related to skin pigmentation. One potential limitation of this study is the relatively low number of Hispanic and sun-protected cases, which may be why some associations within this group were non-significant after adjustment as compared to non-Hispanic white cases. I understand that these cSCC cases are much less common, so I am not sure this necessarily warrants discussion in the manuscript.

We thank the reviewer for the helpful comments. We agree with the reviewer that one potential limitation of our study may be the relatively low number of Hispanic and sun-protected cases, and we added some text in the Discussion to reflect this limitation, as below:

“We recognize several potential limitations of our study ... Finally, certain analytical strata of our study have relatively few cases (e.g. low number of Hispanic/Latino cases compared to non-Hispanic whites, and low number of sun-protected cases) which may limit our power to detect significant associations in these cases.”

Reviewer #3 (Remarks to the Author):

Summary:

In a novel study, Jorgenson et al examine the roles of genetic ancestry, skin pigmentation, and sun exposure for cSCC risk in Hispanics/Latinos and non-Hispanic Whites in GERA cohort. Using principal components computed from genotyping data as a proxy for ancestry, they found significant associations between the two leading principal components and cSCC. The authors then demonstrate that the estimated effects of ancestry are attenuated after conditioning on genetically predicted skin pigmentation, known cSCC risk loci, and a proxy variable for sun exposure. They also performed stratified analysis by anatomical sites of cSCC and a similar analysis in ncSCC to demonstrate the heterogeneous effect of genetic ancestries on different cSCC types. The results were clearly presented and organized in way that made it straightforward to read and interpret. With that said, I have a few comments the authors should address.

We thank the reviewer for the constructive comments.

Major Comments:

1. Several main analyses rely on the use of predicted skin pigmentation from a specific model. I certainly appreciate the fact that the tool has been successfully published and described previously, it would lend much more impact to the current paper if it is at all possible to perform some validation of the predicted pigmentation in the GERA cohort.

We agree with the reviewer that including information on measured skin pigmentation would strengthen our findings, but, unfortunately, we do not have available information on pigmentation traits in the GERA cohort. We have now reflected this point as a potential limitation in the Discussion, as below:

“We recognize several potential limitations of our study ... Second, the skin pigmentation traits that we used in the current study were based on HIrisPlex-S probabilities estimated using genetic information, as direct pigment measures are not available in the GERA cohort. This may result in underestimates of the effects of pigment variation due to phenotype misclassification. However, HIrisPlex-S system has been shown to predict pigment traits accurately when using all the required SNPs (36 for skin color), with average prediction accuracy of 91% for skin color.”

2. Throughout the article the authors successively build upon models to demonstrate the weakened effect of ancestry on cSCC risk. While some models make clear improvements, others are more uncertain. I appreciate the inclusion of p-values for the effect sizes at each stage, however, the manuscript would be significantly strengthened with a formal model comparison analysis. That is, as model complexity increases, can the authors demonstrate that the more complex model is warranted with a likelihood ratio test (or equivalent procedure)? The stronger the evidence under this procedure, the more likely that the component of risk explained by global ancestry is not fully mediated by the additional variables.

As suggested by the reviewer, we now include formal model comparison results using likelihood-ratio tests to determine whether the inclusion of additional variables significantly improves the model fit for these analyses. We found that Model 4 was a significant improvement over the other models. We have added some text to the Methods section (as below) and also reported the likelihood-ratio tests results in a Supplementary Table (Supplementary Table 13):

“The proportion of variance explained by each covariate (i.e. genetic ancestry PCs, HIRISplex-S skin pigment probabilities, GRS based on 14 known cSCC-associated SNPs, or history of AK) and for each ethnic group (i.e. non-Hispanic white or Hispanic/Latino) are reported in Supplementary Table 12. We also conducted formal model comparison analyses (i.e. likelihood-ratio tests) that demonstrate that the more complex model (Model 4) is warranted over the other models (Supplementary Table 13).”

	Model 1 vs. 2	Model 2 vs. 3	Model 3 vs. 4
Added variables	HirisPlex-S skin pigmentation probabilities	GRS (based on 14 known cSCC variants)	Sun exposure (history of AK)
GERA Ethnic Group	Likelihood Ratio Test, P-value		
non-Hispanic white	2.96×10^{-188}	5.73×10^{-54}	$< 2.23 \times 10^{-308}$
Hispanic/Latino	5.58×10^{-17}	1.35×10^{-5}	7.46×10^{-29}

3. The authors report cSCC GWAS results conditional on 14 known risk SNPs. These 14 known SNPs were identified in non-Hispanic whites, and it isn't clear whether the resulting conditional findings claimed to be independent associations are truly independent, or just the result of impartial tagging of the underlying causal variants. That is, if the “known markers” are tagging causal variants, they will be less effective tag SNPs in a Hispanic population with different linkage disequilibrium patterns. Then, a conditional analysis will result in false positive “independent” hits.

To remedy this, I suggest the authors simply perform a statistical fine-mapping at the risk regions in their own data, to identify what likely is causal. This will more clearly reflect what variants are contributing to risk.

This is a good point. To identify which genetic variants within the 18 cSCC- and/or skin pigmentation (14 cSCC + 4 conditional hits at known skin pigmentation loci) are likely to be causal, we used a Bayesian approach (CAVIARBF). We applied this approach on the Hispanic/Latino data and non-Hispanic white data separately for comparison. We have added these results in the manuscript as below, as well as Supplementary Data:

In the Results:

“SNP prioritization and annotations

To identify which genetic variants within the 18 cSCC- and/or skin pigmentation (14 cSCC + 4 conditional hits at known skin pigmentation loci) are likely to be causal, we computed each variant's ability to explain the observed signal and derived the smallest set of variants that included the causal variant with 95% probability. In each of the 18 loci, the corresponding 18 credible sets contained from 1 to 3,868 variants in Hispanic/Latinos (7,391 total variants, Supplementary Data 5); and from 1 to 1,104 variants in non-Hispanic whites (1,373 total variants, Supplementary Data 6). Interestingly, two variants (i.e. IRF4 rs12203592 and SLC45A2 rs16891982) had >50% probability of being causal in both Hispanic/Latinos and non-Hispanic whites. Additional 5 variants (i.e. TYR rs1126809, RALY rs6059655, BNC2-CNTLN rs74664507, DEF8 rs4268748, and HERC2 rs12916300) had >50% probability of being causal in non-Hispanic whites.”

In the Methods section:

“Variants prioritization. To identify which genetic variants within the 18 cSCC- and/or skin pigmentation (14 cSCC + 4 conditional hits at known skin pigmentation loci) are likely to be causal, we used a Bayesian approach (CAVIARBF). For each of the 18 loci, we computed each variant’s capacity to explain the identified signal within a 2 Mb window (± 1.0 Mb with respect to the original top variant) and derived the smallest set of variants that included the causal variant with 95% probability (95% credible set). A total of 7,391 variants within 265 annotated genes were included in these 18 credible sets for the Hispanic/Latino analysis (Supplementary Data 5) and 1,374 variants within 45 annotated genes were included in these 18 credible sets for the non-Hispanic white analysis (Supplementary Data 6).”

4. There is currently not enough information on the models used to test for association between SNP and cSCC risk. This is crucial because is not clear whether population structure was sufficiently controlled.

We have now added additional text to the Methods section providing more information on the models and adjustment for population structure, as follows:

“**Defining genetic ancestry.** The genetic structure of GERA participants has been already characterized using genome-wide genotypes and principal components (PCs) analysis and identified 10 and 6 ancestry PCs reflecting genetic ancestry among non-Hispanic whites, and the other ethnic groups, respectively. In this study, genetic ancestry PCs were calculated by Eigenstrat v4.2. In the current study, for genetic ancestry analyses, for each ethnic group, we focused on PC1 and PC2 because they are the only PCs that are geographically interpretable. In non-Hispanic whites, PC1 represents a European northwestern vs. southeastern cline, and PC2 represents a European northeastern vs. southwestern cline. In Hispanic/Latinos, PC1 represents greater European versus Native American ancestry and PC2 represents greater African versus European ancestry.”

“**Statistical analysis.** We used a logistic regression model to examine the impact of ancestry on cSCC risk using R version 3.4.1 with the following covariates: age, sex, and ancestry PCs (Model 1). For all the analyses, we included as covariates the top 10 ancestry PCs for the non-Hispanic whites, whereas we included the top 6 ancestry PCs for the Hispanic/Latinos. To adjust for genetic ancestry, we also included the percentage of Ashkenazi (ASHK) Jewish ancestry as a covariate for the non-Hispanic white ethnic group analysis. We present the full multivariate logistic regression models of cSCC and ncSCC in Supplementary Tables 1-2, and Supplementary Tables 6 and 8, including all the PCs for each ethnic group, and adjusting for age and sex.”

5. It would be interesting to see the effect of local ancestry at both the known cSCC sites and pigmentation sites in the GERA cohort. That is, for the Hispanic individuals, can the authors call local ancestries (using RFMix or similar tool) and perform a localized admixture scan? These results would help bolster the currently presented findings.

This is an excellent suggestion. We have now conducted local ancestry analyses in GERA Hispanic/Latinos using a method for local ancestry inference which leverage the structure of linkage disequilibrium in the ancestral population (LAMP-LD). Interestingly, we detected a significant association between cSCC risk and local European genetic ancestry at **SLC24A5**, where on average cases were of 91.8% European descent compared with controls at 3.8% ($P=1.71 \times 10^{-4}$). We have added some text through the manuscript to reflect these new findings:

In the Results section:

“**Local ancestry**

To determine whether each of the 16 known cSCC- and/or skin pigmentation-associated loci show evidence of increased European ancestry in cSCC cases compared to controls, we conducted local ancestry analyses in GERA Hispanic/Latinos using a method for local ancestry inference which leverage the structure of linkage disequilibrium in the ancestral population (LAMP-LD). After correction for testing multiple loci, we detected a significant association between cSCC risk and local European genetic ancestry at **SLC24A5**, where on average

cases were of 91.8% European descent compared with controls at 3.8% ($P=1.71\times 10^{-4}$) (Supplementary Table 7 and Supplementary Figure 3).”

In the Methods section:

“Local ancestry

To determine whether the proportion of European ancestry at the 16 known cSCC- and/or skin pigmentation-associated loci could account for cSCC risk in Hispanic/Latinos, we conducted local ancestry analyses using LAMP-LD. We used the "HGDP-CEPH Human Genome Diversity Cell Line Panel" (<https://hagsc.org/hgdp/>). Association testing between cSCC status and percent European ancestry at each locus was performed using the nonparametric test statistic proposed by Montana and Pritchard for admixture mapping. For this analysis, a Bonferroni-corrected significance threshold of $0.05/16 = 3.1\times 10^{-3}$ (accounting for the number of loci tested) was applied. We used the R package “ggplot2” to generate the local ancestry at **SLC24A5** plot presented in Supplementary Figure 3.”

In the Discussion:

“We also found evidence that the percentage of European ancestry at **SLC24A5** was strongly correlated with cSCC case status ($P=1.71\times 10^{-4}$) in Hispanic/Latinos. cSCC cases contained on average 91.8% European ancestral alleles at the **SLC24A5** locus compared to 3.8% in controls, suggesting that European-specific variation may contribute to cSCC risk at this locus. **SLC24A5** is a well-known target of positive selection, associated with lighter skin color in admixed populations. Indeed, we showed that the well known amino acid substitution at this locus, Ala111Thr, previously shown to be continentally dichotomous between Europeans and the rest of the world, and strongly associated with pigment in populations in which it is polymorphic (e.g. Hispanics and South Asians), is also a strong risk factor for cSCC in our Hispanic/Latino subjects. This is an important novel cSCC association uniquely found in this population, to date. Future studies can investigate whether this variant may also be an important risk factor in African American populations.”

Minor Comments:

1. The gradient figures are very nice and incredibly helpful to see the geographic/ancestral distribution of variables. It be helpful to use a more colorblind-friendly palette (e.g., viridis).

As suggested, we have now generated all the geographic/ancestral gradient plots as viridis-based plots and provide those a separate Supplemental Material.

2. The authors concluded (p.12) the variation in cSCC risk within Hispanic/Latino and non-Hispanic white populations “is due in part to skin pigmentation genes and the interaction of those genes with sun exposure leading to higher risk.” I would consider rewording this statement as the only models investigated were additive with respect to risk. An interesting analysis would be an interaction between predicted pigmentation and sun exposure.

We have reworded this statement in the Discussion to reflect that the analyses were focused on additive effects, as below:

“This variation in cSCC risk is due in part to genetic variation in skin pigmentation genes and its interaction with sun exposure leading to higher risk. These results, based on additive models, underscore the importance of incorporating genetic ancestry, skin pigmentation traits, and sun exposure into genetic analyses of skin cancer.”

3. Skin pigmentation were originally classified by the HIrisPlex-S into 5 levels. However, the authors could explain why they regrouped the pigmentation into 3 levels: “dark skin + dark to black skin”, “intermediate skin”, “very pale skin + pale skin”. I wonder how the 5 levels will be correlated with PCs, whether there will be a linear trend in their effects on cSCC, and how much associations between cSCC and PCs would be left when the original scale was used.

We have added some text in the Methods section to justify our choice of presenting a 3-category level (very pale + pale, intermediate, dark + dark-black) of the HIRISplex-S skin pigmentation predictions, as follows:

“We did consider using a 5-category level (very pale, pale, intermediate, dark and dark-black) of the HIRISplex-S skin pigmentation predictions and examined the distribution of all the categories in GERA non-Hispanic whites (Supplementary Figure 6). Because the distribution of the categories “very pale” and “dark to black” were not well-differentiated from “pale” and “dark”, respectively, we decided to combine “dark” and “dark to black skin” into a single category, and “very pale” and “pale” into a single category. Thus, by focusing on a 3-category level (very pale + pale, intermediate, dark + dark-black), we improved the predictive values for skin pigmentation compared to the original 5-category level.”

We have also added a Supplementary Figure (Supplementary Figure 6):

REVIEWERS' COMMENTS:

Reviewer #3 (Remarks to the Author):

The authors performed a considerable amount of work to address my comments [as well as the comments from other reviewers] and I find the manuscript to be much improved as a result.

I have no further comments.